# Epidemiological and Microbiological Aspects of the Peritonsillar Abscess

**DOI:** 10.3390/ijerph17114020

**Published:** 2020-06-05

**Authors:** David Slouka, Jana Hanakova, Tomas Kostlivy, Petr Skopek, Vojtech Kubec, Vaclav Babuska, Ladislav Pecen, Ondřej Topolcan, Radek Kucera

**Affiliations:** 1Department of Otorhinolaryngology, University Hospital in Pilsen, Faculty of Medicine in Pilsen, Charles University, 300 00 Pilsen, Czech Republic; slouka@fnplzen.cz (D.S.); hanakovaj@fnplzen.cz (J.H.); kostlivyt@fnplzen.cz (T.K.); skopekp@fnplzen.cz (P.S.); kubecv@fnplzen.cz (V.K.); 2Department of Medical Chemistry and Biochemistry, Faculty of Medicine in Pilsen, Charles University, 300 00 Pilsen, Czech Republic; vaclav.babuska@lfp.cuni.cz; 3Department of Immunochemistry Diagnostics, University Hospital in Pilsen, Faculty of Medicine in Pilsen, Charles University, 300 00 Pilsen, Czech Republic; ladislav.pecen@seznam.cz (L.P.); topolcan@fnplzen.cz (O.T.)

**Keywords:** peritonsillar abscess, incidence, bacteriology, primary prevention, patient stratification, personalized treatment

## Abstract

Peritonsillar abscess (PTA) is the most common complication of tonsillitis. Cultivation usually reveals a wide spectrum of aerobic and anaerobic microbiota. This retrospective study compared PTA incidence and the spectrum of individual microbial findings in groups of patients divided by gender, age, and season. Of the 966 samples cultivated, a positive cultivation finding was detected in 606 patients (62.73%). Cultivation findings were negative in 360 (37.27%), meaning no pathogen was present or only common microbiota was cultivated. The highest incidence of PTA was found in group I patients (19–50 years) (*p* ≤ 0.0001) and the most frequently cultured pathogens was *Streptococcus pyogenes* (36.23%). Gender seemed to have an influence on the results, with higher incidence found in males (*p* ≤ 0.0001). The analysis of correlation between PTA incidence and season did not yield statistically significant results (*p* = 0.4396) and no statistically significant differences were observed in individual pathogen frequency. PTA had a higher incidence in adult males and a slightly higher incidence in girls in childhood. The following findings are clinically significant and have implications for antibiotic treatment strategy: (1) the most frequently cultivated pathogen was *Streptococcus pyogenes*; (2) an increased incidence of anaerobes was proven in the oldest group (>50 years).

## 1. Introduction:

Peritonsillar abscess (PTA) is one of the most frequent local complications of acute tonsillitis; the incidence of its occurrence in the population is reported as 10–45/100,000 people. [1,2]. It is diagnosed in patients of almost all age groups, most often in adolescents and young adults. In the vast majority of cases PTA is unilateral, but rarely occurs bilaterally [3,4]. The ratio of left- and right-side disabilities is reported to be approximately the same or with a slight predominance of the left side [5,6]. First, a peritonsillar phlegmon develops, then a pyogenic membrane forms, and the inflammatory process obtains the form of abscess [7,8]. PTA may be accompanied by the development of other complications such as airway obstruction [9]. Furthermore, deep throat infections with parapharyngeal, retropharyngeal, and visceral neck spaces, mediastinitis, necrotizing fasciitis, internal carotid artery erosion, and brain abscess may occur. [10,11,12]. The patient may alternate between individual forms of deep neck infection complications. These are always serious, life-threatening conditions, requiring broad interdisciplinary cooperation [13,14]. The mortality rate of mediastinitis is still reported to be 20%–50% despite the wide availability of antibiotic (ATB) therapy [15,16].

The diagnosis of PTA is based on a history combined with a general clinical assessment. A patient with PTA typically presents with fever, a sore throat, unclear speech, sometimes trismus, or a reaction of the descending lymph nodes. Clinically, redness and arching of the palpably stiff and markedly painful soft palate are found [17,18].

The aim of the study was to evaluate PTA incidence and its microbial spectrum in the group of patients as relating to gender, age, and season in order to assess the importance of different variants in ATB therapy management.

## 2. Materials and Methods

### 2.1. Group of Patients

A total of 966 patients were enrolled in this retrospective monocentric study. Informed consent was obtained from all the participants. The study was approved by the Ethical Committee of the University Hospital in Pilsen on 8th July 2013. The cohort includes 396 (41.99%) females and 570 (59%) males who were diagnosed with PTA between the years 2014–2018. The patients’ ages ranged from 2 to 86 years of age, with an average of 36.7 years of age for the whole group. The average age in the female group was 35.9 years. The average age in the male group was 37.4 years. Patient distribution by gender and age is shown in Figure 1. 

Inclusion criteria were setup as follows: diagnosis of PTA confirmed by perioperative findings, tonsillectomy under general anesthesia, and histological findings from perioperatively collected material confirming PTA. Exclusion criteria included: peritonsillar phlegmon, outpatient treatment, and failure to perform tonsillectomy.

The initiation of ATB therapy was not an exclusion criterion. Our current study reflects the real spectrum of patients with the diagnosis of peritonsillar abscess treated at the University Hospital.

The relationship between the spectrum of microbial agents and gender, age and season was evaluated in all the groups. Age groups were created as follows: 0–18 years, 19–50 years, >50 years. Three groups were created based on knowledge gained through clinical practice. We have observed, throughout our practice, a different course of infection and reaction to treatment, as well as differences in the prevalent microbiota in children and teenagers (group 0–18 years), adults (group 19–50 years) and older adults (group >50 years). The age and gender characteristics of the patient group are given in Table 1. The seasons were defined as follows: spring from the 1st of March to May 31st, summer from the 1st of June to the 31st of August, autumn from the 1st of September to November 30th and winter from the 1st of December to the 28/29th of February.

### 2.2. Methods Used

A material for microbiological examination was obtained in all patients perioperatively. The samples contained specimens of affected tonsil tissue and pus from the PTA by aspirating using a sterile syringe. The samples were processed at the Department of Microbiology, University Hospital in Pilsen within 24 h. The material was cultivated parallel, aerobically, and anaerobically. Aerobic cultivation was performed on blood agar and Endo agar in a thermostat at 37 °C for 48 h. Schaedler agar was used for the anaerobic cultivation, which was performed in an anaerobic box for 72 h. Propagation and inoculation were performed simultaneously on solid culture media: the material was inoculated into a liquid anaerobic propagation medium (Wilkins-Chalgren broth), which was cultivated in an anaerobic box for 48 h, then the broth containing propagated microbes was inoculated again on solid media: blood agar and Schaedler agar. MRSA was determined using an oxacillin-sensitivity test in special plates [19]. We have taken note of the common oropharyngeal microbiota as described by Aas at al. [20]. Individual microbial agents which were part of the pathogenic microbiota are shown in Table 2, Table 3 and Table 4.

### 2.3. Statistical Methods

A statistical analysis was performed using SAS software (Statistical Analysis Software release 9.4; SAS Institute Inc., Carry, NC, USA). A comparison of the groups of patients was performed using a chi-square or Fisher’s exact test categorical parameters if the expected frequency in any subgroup was less than five patients. For continuous variables, subgroup comparisons were performed using the Wilcoxon (two subgroups compared) or Kruskal–Wallis test (more than two subgroups compared). For all hypotheses tested, a *p*-value less than 0.05 meant statistical significance. All tests were performed as two-side tests. For better orientation, some results are also presented in graphic form.

## 3. Results

Of the 966 patient samples examined, a positive cultivation finding (at least one pathogen) was detected in 606 patients (62.73%). In 360 (37.27%), the cultivation finding was negative, i.e., no pathogen was present, or only common oropharyngeal microbiota was cultivated. Of the microbial agents, aerobes (*n* = 557, 75.77%), anaerobes (*n* = 171, 22.95%), and yeast (*n* = 17, 2.3%) were found in our group. The most frequently cultivated pathogen was *Streptococcus pyogenes* (*n* = 351, 36.34%).s

The evaluation of the frequency of microbial agents in each gender is summarized in Table 2. In general, males had a numerical predominance over females (*p* ≤ 0.0001). A statistically significant higher incidence of aerobes (*p* ≤ 0.0001) and anaerobes (*p* ≤ 0.0001) was observed in males. Positive cultivations in the male group included: aerobes—343 patients (60.18%), anaerobes—118 patients (20.70%), and yeasts—11 patients (1.93%). Positive cultivations in the female group included: aerobes—213 patients (53.79%), anaerobes—52 patients (13.13%), and yeasts—6 patients (1.51%). Only *Fusobacterium species* showed statistically significant differences for individual agents (*p* = 0.0078).

Evaluation of the frequency of microbial agents in the different age groups is summarized in Table 3. The highest incidence of PTA was in the 19–50 years age group—614 patients (*p* ≤ 0.0001). In the 0–18 age group aerobes—63 patients (45.00%) and anaerobes—18 patients (12.86%) were positively cultivated; yeasts were not detected. The most frequently cultivated pathogen was *Streptococcus pyogenes*—31 patients (22.14%). In the 19–50 years age group aerobes—377 patients (61.40%), anaerobes—86 patients (14.00%), and yeasts—9 patients (1.47%) were positively cultivated. The most frequent pathogen was *Streptococcus pyogenes*—158 patients (25.73%). In the >50 years group aerobes—116 patients (54.72%), anaerobes—67 patients (31.60%), and yeasts—8 patients (3.77%) were positively cultivated. The highest increase of incidence was observed in anaerobic agents in the >51 years age group (31.60%), compared to both younger groups (12.86%, 14.00%) (*p* < 0.0001). Statistically significant differences across the age groups were observed between a number of individual agents in all three groups (aerobes, anaerobes, and yeasts). The most common pathogens were streptococci (other species), which were present in 40 patients (18.87%).

The impact of season on the prevalence of different microbial agents is summarized in Table 4. PTA incidence was not found to be dependent on the season in any statistically significant way (*p* = 0.4396). Evaluating the distribution of microbial agents within seasons, we can state that season dependence has not been proven. Season does not seem to impact whether the most prevalent pathogens are aerobes, anaerobes, or yeasts, nor does it have an effect on the different types of pathogens found.

When we examined the obtained results more closely and combined the setup criteria (gender, age, and season), we found the following: in the 0–18 age group, girls had a higher incidence of PTA, but without a statistical significance (*n* = 76, 54.3%, *p* = 0.3105). Among patients in the 19–50 age group, males had a higher incidence of PTA (*n* = 378, 61.6%, *p* ≤ 0.0001). The >50 age group presented a higher incidence of PTA in males (*n* = 128, 60.4%, *p* = 0.0025). The highest incidence of PTA in boys in the 0–18 age group was in the summer and the lowest was in the autumn (*p* = 0.0023). Among females, the >50 age group had the highest incidence of PTA in the winter and a significantly lower incidence in the summer and autumn (*p* = 0.0284). The most common pathogen in the 19–50 age group, *Streptococcus pyogenes,* has a statistically significant higher incidence in the summer and is at its lowest in the spring (*p* = 0.0207), *Staphylococcus aureus* in the 0–18 age group has the highest incidence in the summer and is least prevalent in the autumn (*p* = 0.0192).

## 4. Discussion

Despite advanced ATB therapy, PTA remains a serious and, if allowed to progress, life-threatening complication, of acute tonsillitis that affects a broad age range of patients. The considerable amount of attention PTA receives in studies is therefore justified.

### 4.1. Gender and Age Spectrum in the PTA Patient Group

Published studies report approximately the same incidence of PTA in both genders [21] or a slight predominance in males in the adult population [22]. In childhood a slight predominance is described in girls [6]. In our group, regardless of age, males had a statistically significant predominance (*n* = 570, 59.0%, *p* < 0.0001). After stratifying the patients by age (Table 1), as well as in accordance with the literature on adults, males had a statistically significant predominance in groups II and III (*n* = 378, 61.6%, *p* < 0.0001 and *n* = 128, 60.4%, *p* = 0.0025). In group I, of children under 18 years of age, girls slightly predominated (*n* = 76, 54.3%, *p* = 0.3105), but in this case the difference was not statistically significant. The mean age of patients with PTA in our cohort was 36.7 years of age. The highest incidence of PTA occurs between 15 and 35 years of age and is reported in the available literature. In our group, the highest incidence of PTA was in age group II (19–50 years), 614 patients (63.56%). Group I (18 years of age, or younger) had 140 patients with PTA (14.49%) and group III (>50 years) 212 patients (21.95%), which corresponds to data in the literature [1,2,23].

### 4.2. The Spectrum of Microbial Agents

The infectious bacterial agents involved in PTA pathology may vary according to different clinical factors. The most frequently mentioned cultivation findings in the literature are aerobic, namely β-hemolytic streptococci, streptococci of other groups, then *Haemophilus influenzae, Staphylococcus aureus, Klebsiella pneumoniae,* and *Enterobacter species a Pseudomonas aeruginosa* [7,24]. Of the anaerobes, *Fusobacterium species, Peptostreptococcus species, Bacteroides species,* and *Prevotella species* were most predominate. Yeasts were rarely cultivated [25,26,27]. In our results, in agreement with the large number of published studies on the topic [2,3,5,7,12,25,28], aerobic cultivation findings predominated (*n* = 557, 75.77%) and the most frequently cultured pathogen was *Streptococcus pyogenes* (*n* = 351, 36.34%). However, this contradicts the work of Prior et al., who pointed out the predominance of anaerobic bacteria [29]. Prior et al. found the presence of anaerobes in 84% of cultivations. However, their work was limited by a small group of patients (*n* = 53).

Insofar as successful cultivation is concerned, in our group of patients, pathogens were successfully cultivated in 606 cases (62.73%). In the remaining 360 cases (37.27%), the cultivation finding was negative, or contained only normal oropharyngeal microflora, which is in line with data presented by a number of other studies [1,6,13,23,24,25,29]. Whether or not bacteria are cultivated successfully can be affected by a number of factors. The most significant factor is the initiation of ATB therapy before the collection of the material for cultivation. In the cohort we presented, there were 386 patients who used ATB before the sample for cultivation was collected. No pathogen was cultivated in this group in 297 cases (76.94%). The effect of ATB therapy on the result of cultivation is therefore clear from our results. However, Love et al. reported an unclear effect of ATBs use prior to material collection on cultivation results [30]. In their group, 39% of patients used ATB before the samples for cultivation were collected and the cultivation results did not differ in any way from the results of patients without previous ATB treatment, which is in conflict with our results.

### 4.3. The Spectrum of Microbial Agents based on Gender, Age and Seasons

The broad spectrum of the bacterial microbiota associated with PTA and the age dependent changes it undergoes have a multifactorial origin. The spaces of the oral cavity and cervical tonsils are both heavily, yet very differently, colonized. Bacteria that are thought to have pathogenic potential may occur in healthy individuals [31,32]. The difference in the bacterial spectrum of tonsillitis in childhood and adulthood is documented [33]. The impact of social history, climate and local ATB policy on the bacterial population of the tonsils and thus their impact on the current bacterial spectrum of PTA has to be taken into account [34,35].

The increased incidence of anaerobic agents in group III compared to I and II (31.64% vs. 14.1% and 12.86%) has a significant clinical impact on our results. In agreement with other authors, we do not attach clinical significance to the absence of yeast in group I (18 years or younger age group) compared to the culture results in both older groups (1.47% and 3.77%) [13,23]. We view the absence of yeast in group I (18 years or younger) and the higher proportion of the anaerobic spectrum in elderly patients as a result of higher incidence of metabolic and cardiopulmonary comorbidities in elderly patients. This is in concordance with Gavriel et al. [36]. Differences in the PTA incidence in individual periods of the year (spring, summer, autumn, and winter) are not significant in our group (*p* = 0.4396), nor were they significant in studies from Denmark, England, or the USA [13,23,37]. Other studies have reported significant differences in the seasonal incidence of PTA in a group of pediatric patients with the highest incidence in the spring or summer [6,38,39]. In our work, a detailed analysis revealed that the incidence of PTA in boys in group I (0–18 years) is statistically significantly higher during the summer and, conversely, is at its lowest in the autumn (*p* = 0.0023).

If we take age into account, it is possible to trace partial seasonal effects on the frequency of individual agents. For example, in group II (19–50 age group), *Streptococcus pyogenes* has the highest statistically significant incidence in the summer and the lowest in the spring (*p* = 0.0207). In group I (0–18 age group) *Staphylococcus aureus* has the highest incidence in the summer, the lowest in the autumn (*p* = 0.0192). In agreement with the literature [1,2,6,23,38,39], we consider the findings regarding the seasonality of the individual microbial agents interesting from an epidemiological point of view. From a clinical point of view, the treatment strategy remains the same.

### 4.4. The Spectrum of Microbial Agents and PTA Therapy

The basis of successful PTA therapy is an early diagnosis and initiation of abscess drainage with concomitant empirical administration of ATB [25,40,41]. Tonsillectomy on the day of admission to hospital is now considered a standard treatment, unless there are contraindications to the procedure [28,42]. The ATB of first choice is parenterally administered penicillin [5,7]. In case of allergy to penicillin ATBs, a replacement from the macrolide or lincosamide group is chosen [23,30]. The preferred ATB scheme varies from country to country. In the work of Wikstein et al. [2], which describes treatment in the Nordic countries of Europe (Denmark, Norway, Sweden, and Finland), penicillin was the preferred first choice in 65% of cases. In contrast, separate Danish and UK studies preferred a combination of penicillin and metronidazole treatment in approximately 60% of cases [13,43]. Other studies yet present the combination of potentiated amoxicillin with metronidazole [44], cefuroxime with metronidazole [45], or the administration of clindamycin alone [13]. What all these studies have in common is that they reflect the development of PTA bacterial findings. Previously the genus *Streptococcus* was considered predominant in PTA associated microbiota, whereas today a significant proportion of anaerobic microbiota is also taken into account [27,46].

The results of our extensive cohort are in line with these trends and offer conclusions that can help personalize ATB therapy. Our results for groups I and II (patients <50 years of age), where *Streptococcus pyogenes* was present most frequently, validate the preference of penicillin as a first-choice ATB in accordance with the scheme published by Wikstein et al. [2]. Furthermore, a significant decrease in aerobic agents alongside an increased incidence of anaerobic agents in the oldest group, group III. (patients >50 years), fully supports the combination of penicillin with metronidazole in older patients [44]. If anamnestic data indicates an allergy to penicillin antibiotics, we prefer a replacement from the macrolide or lincosamide groups [13,47].

## 5. Conclusions

PTA is the most common complication of tonsillitis. We found that PTA has a higher incidence in males in adulthood and a slightly higher incidence in girls in childhood. Aerobic agents predominated in the cultivation findings and the most frequently cultivated pathogen was *Streptococcus pyogenes.* Examining the relationship between microbial agents and age, we found an increased incidence of anaerobes in the oldest group (over 50 years) and an absence of yeasts in the youngest group (18 years of age or younger). The occurrence of many specific microbial agents was age related. We did not prove a dependence of PTA incidence on the season. The current methodology in the personalization of ATB therapy is consistent with our microbiology results. Penicillin is the first-choice ATB in patients under the age of 50. In older age groups an increasing probability of the presence of anaerobes must be taken into account and the therapy should be modified accordingly. A certain limitation of our study may be its monocentricity. A significant advantage however is the high number of monitored patients (*n* = 966) in a relatively short period of 4 years.

## Figures and Tables

**Figure 1 ijerph-17-04020-f001:**
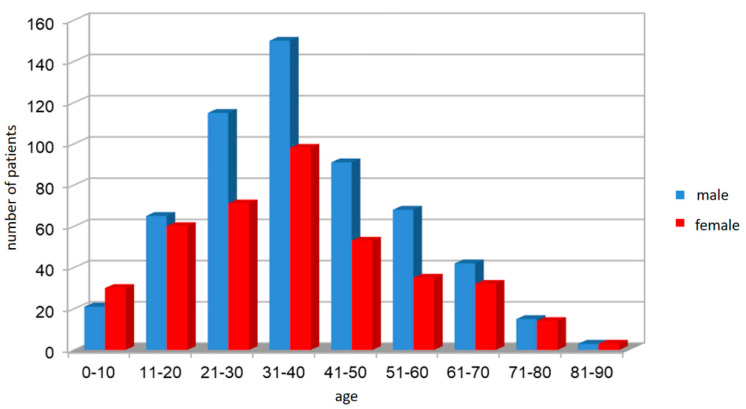
Age and gender distribution of patients.

**Table 1 ijerph-17-04020-t001:** Age and gender characteristics of the patient group.

Age Category	Male (*n*/%)	Female (*n*/%)	Total (*n*/%)	M/W Difference *p*-Value
I. 0–18	64/ 45.7	76/ 54.3	140/ 14.5	0.3105
II. 19–50	378/ 61.6	236/ 38.4	614/ 63.6	<0.0001
III. >50	128/ 60.4	84/ 39.6	212/ 21.9	0.0025
Total (*n*/%)	570/ 59.0	396/ 41.0	966/ 100	<0.0001

**Table 2 ijerph-17-04020-t002:** The spectrum of microbial agents based on gender.

Group	Male	Entire Group	Female	Entire Group	*p*-Value
Incidence	N	%	N	%	
Total (*n* = 966)	570	59.01%	396	40.99%	<0.0001
**Microbial agents**	**n**	**Group Male (%)**	**n**	**Group Female (%)**	
Negative cultivation result	98	17.19	125	31.57	-
Streptococcus pyogenes	128	22.46	82	20.71	0.4551
Streptococcus (other species)	93	16.31	47	11.87	0.1145
Staphylococcus aureus	63	11.05	51	12.88	0.3644
MRSA	2	0.35	3	0.75	0.4286
Escherichia coli	5	0.88	2	0.50	0.4531
Serratia marcescens	3	0.53	1	0.25	0.4771
Pseudomonas aeruginosa	2	0.35	0	0	0.2232
Haemophilus influenzae	9	1.58	8	2.02	0.7039
Klebsiella pneumoniae	17	2.99	9	2.28	0.6699
Enterobacter species	20	3.51	9	2.28	0.2034
Granulicatella species	1	0.18	0	0	0.3893
Gemella morbillorum	0	0	1	0.25	0.2449
Aerobes total	343	60.18	213	53.79	<0.0001
Fusobacterium species	45	7.88	15	3.79	0.0078
Prevotella species	36	6.32	19	4.80	0.3404
Veillonella species	26	4.56	12	3.03	0.1634
Peptostreptococcus species	11	1.93	4	1.01	0.2102
Actinomyces species	0	0	1	0.25	0.2449
Bacteroides species	0	0	1	0.25	0.2449
Anaerobes total	118	20.70	52	13.13	<0.0001
Candida albicans	11	1.93	6	1.51	0.2253

Legends: Table 2 shows PTA patient incidence (N) in both gender groups. Percentage of incidence in each gender group was calculated from the total number of patients (966). Next, the individual microbial agents cultivated (n) in the different gender groups are presented. The percentage of the cultivated individual agents in each gender group was calculated from the total number of patients (N) in the group. For better orientation, the microbial agents were color coded according to their kind: aerobes: blue; anaerobes: yellow; yeast: green. MRSA is methicillin-resistant *Staphylococcus aureus*.

**Table 3 ijerph-17-04020-t003:** The spectrum of microbial agents based on age.

Age Incidence	0–18		19–50		>51		*p*-Value
	N	Entire Group (%)	N	Entire Group (%)	N	Entire Group (%)	
**Total (*n* = 966)**	140	14.49	614	63.56	212	21.95	<0.0001
**Microbial agents**	**n**	**Group 0–18** **(%)**	**N**	**Group 19–50 (%)**	**N**	**Group >51** **(%)**	
Negative cultivation result	59	42.14	142	23.12	21	9.91	-
Streptococcus pyogenes	31	22.14	158	25.73	21	9.91	0.0001
Streptococcus (other species)	12	8.57	88	14.33	40	18.87	0.0267
Staphylococcus aureus	11	7.87	81	13.19	22	10.38	0.1615
MRSA	4	2.86	1	0.16	0	0	0.0002
Escherichia coli	1	0.71	3	0.49	3	1.42	0.3905
Serratia marcescens	0	0	2	0.33	2	0.93	0.3432
Pseudomonas aeruginosa	0	0	0	0	2	0.93	0.0283
Haemophilus influenzae	1	0.71	12	1.95	4	1.89	0.5947
Klebsiella pneumoniae	0	0	17	2.77	9	4.25	0.0539
Enterobacter species	2	1.43	14	2.28	13	6.13	0.0090
Granulicatella species	1	0.71	0	0	0	0	0.0522
Gemella morbillorum	0	0	1	0.16	0	0	0.7506
Aerobes total	63	45.00	377	61.40	116	54.72	<0.0001
Fusobacterium species	6	4.29	34	5.54	21	9.91	0.0446
Prevotella species	5	3.57	28	4.57	22	10.38	0.0035
Veillonella species	5	3.57	16	2.61	17	8.02	0.0022
Peptostreptococcus species	2	1.43	7	1.14	6	2.83	0.2275
Actinomyces species	0	0	0	0	1	0.47	0.1686
Bacteroides species	0	0	1	0.16	0	0	0.7506
Anaerobes total	18	12.86	86	14.00	67	31.60	<0.0001
Candida albicans	0	0	9	1.48	8	3.77	0.0204

Legends: Table 3 shows the PTA patient incidence (N) in the different age groups. The percentage of incidence in each age group was calculated from the total number of patients (966). Next, the individual microbial agents cultivated (n) in the different age groups are presented. Percentage of the cultivated individual agents in each age group was calculated from the total number of patients (N) in the group. For better orientation, the microbial agents were color coded according to their kind: aerobes: blue; anaerobes: yellow; yeast: green. MRSA is methicillin-resistant *Staphylococcus aureus*.

**Table 4 ijerph-17-04020-t004:** The spectrum of microbial agents based on seasons.

Seasons	Spring		Summer		Autumn		Winter		*p*-Value
Incidence	N	Entire Group (%)	N	Entire Group (%)	N	Entire Group (%)	N	Entire Group (%)	
**Total (*n* = 966)**	246	25.47	259	26.81	224	23.19	237	24.53	0.4396
**Microbial Agents**	**n**	**Spring Period (%)**	**n**	**Summer Period (%)**	**n**	**Autumn Period (%)**	**n**	**Winter Period (%)**	
Negative cultivation result	61	24.80	71	27.41	15	6.70	75	31.65	-
Streptococcus pyogenes	48	19.51	46	17.76	61	27.23	55	23.21	0.4012
Streptococcus (other species)	33	13.41	32	12.36	36	16.07	39	16.46	0.8282
Staphylococcus aureus	31	12.60	28	10.81	31	13.84	24	10.13	0.7549
MRSA	0	0	0	0	4	1.78	1	0.42	0.0349
Escherichia coli	0	0	2	0.77	4	1.78	1	0.42	0.1712
Serratia marcescens	0	0	2	0.77	2	0.89	0	0	0.2610
Pseudomonas aeruginosa	0	0	1	0.39	1	0.45	0	0	0.5722
Haemophilus influenzae	4	1.62	5	1.93	4	1.78	4	1.70	0.8803
Klebsiella pneumoniae	7	2.85	6	2.32	10	4.45	3	1.26	0.2757
Enterobacter species	7	2.85	13	5.02	6	2.68	3	1.26	0.0621
Granulicatella species	0	0	0	0	1	0.45	0	0	0.3915
Gemella morbillorum	0	0	0	0	1	0.45	0	0	0.3915
Aerobes total	130	52.85	135	52.12	161	71.88	130	54.85	0.1900
Fusobacterium species	18	7.32	13	5.02	15	6.70	15	6.33	0.8377
Prevotella species	13	5.28	19	7.34	15	6.70	8	3.38	0.2012
Veillonella species	11	4.47	13	5.02	11	4.91	3	1.26	0.0992
Peptostreptococcus species	5	2.03	4	1.54	3	1.36	3	1.26	0.8647
Actinomyces species	1	0.41	0	0	0	0	0	0	0.3915
Bacteroides species	1	0.41	0	0	0	0	0	0	0.3915
Anaerobes total	49	19.91	49	18.92	44	19.64	29	12.24	0.0985
Candida albicans	6	2.44	4	1.54	4	1.78	3	1.26	0.7728

Legends: Table 4 shows PTA patient incidence (N) in the different seasons. Percentage of incidence in each season was calculated from the total number of patients (966). Next, the individual microbial agents cultivated (n) in the different seasons are presented. The percentage of the cultivated individual agents in each season was calculated from the total number of patients (N) in the season. For better orientation, the microbial agents are color coded according to their kind: aerobes- blue, anaerobes- yellow, yeast- green. MRSA is methicillin-resistant *Staphylococcus aureus*.

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
