# Peer review of "Epidemiological and Microbiological Aspects of the Peritonsillar Abscess"

_ijerph, 2020, doi:10.3390/ijerph17114020_

Round 1

Reviewer 1 Report

It is a retrospective analysis of a very large series of patients with peritonsillar abscess attended in a single institution in 4 years. All patients were submitted to culture and tonsillectomy

Inclusion and exclusion criteria should be presented at the begining of the Methods; There is no reference in these criteria about complication of the PTA as parapharyngeal or retropharyngeal abscess. 

Gender should be refered as male and female instead to women and men.

The legend in Figure 1 are not written in English.

In the inclusion criteria and results there is no reference to the previous use of antibiotics. It is very important to analyze the positive culture and the microbial spectrum. In addition, there is no reference to antibiotic resistance in culture (antibiogram). 

Table 2 and Figure 2 present the same information. The same occurs in table 3 / figure 3, and table 4 / figure 4. 

Evolution of the infection was not presented. Although it is not the aim of the study, It will be interesting to correlate isolated agent and evolution of the infection. Also it will be important to support antibiotical therapy choice.

Author Response

                                                                              Pilsen, 27th May, 2020

Dear Reviewer 1,

Thank you for your comments to the article: "Epidemiological and Microbiological Aspects of The Peritonsillar Abscess

We went carefully through your comments. Please, find below our answers to each comment.

We also improved the level of English. Article was checked by a native speaker with biology education, focused on microbiology.

We believe that all the changes done increased the level of the article.

Kind regards,

David Slouka

Reviewer 2 Report

In this manuscript microbiological and epidemiological aspects of Peritonsillar Abscess are studied. Microbiological characteristics of the studied population are described.

The following comments are attached:

  1. Lines 18-19: The phrase is not clear, you says that 606 samples (67.73%) were positive and “The remaining results were divided between common oropharyngeal microflora in 360 patients, or a completely negative cultivation”. It is not understood how many have a common microbiota and how many are negative. Besides correcting all the microflora text, the correct thing is microbiota.
  2. Figure 1. It can be eliminated since it does not represent the studied groups. If you decide to leave it, you must translate into English all the texts in the figure.
  3. Line 55: You do not explain the criteria by which you only divided the patients into three groups.
  4. Line 73: You do not explain within the criteria the taking of antibiotics in the patients. Was it an exclusion criterion that the patients were taking antibiotics? If it was an exclusion criterion, how long after taking antibiotics was the sample taken?
  5. Line 75: The methodology used is not adequately described. How were the samples taken and from which site? How did you identify the genus and species of the isolated bacteria, what methodology did you use? How did you determine the MRSA strains? It is recommended to make a better description of the microbiological methodology used.
  6. Line 80: You do not explain what was the purpose of the propagation. What conditions were used in these samples for anaerobiosis?
  7. Line 94: What are the bacteria that you considered pathogenic? What bacteria did you consider a common microbiota?
  8. Table 2: What does “Negative cultivation result” mean? Didn't bacteria grow or do you have pathogenic bacteria? Indicate what MRSA means.
  9. Figure 2: You can be eliminated does not provide more information from Table 2, if you decide to leave it you must put the name of the bacteria in italics, without abbreviating and using upper and lower case.
  10. Figure 3. Same comments as for Figure 2.
  11. Line 139: Why did you suppose there could be a change according to the seasons?
  12. Figure 4. Same comments as for Figure 2.
  13. Line 191: Were the growth conditions of Prior et al. similar to yours?
  14. Line 196: The correct thing is microbiota not microflora.
  15. Lines 199-200: There is no mention in Materials and Methods regarding antibiotics, you do not mention which antibiotics were used by patients or how long after their use was the taking of samples.
  16. Line 207: Must say microbiota
  17. Line 215: Delete i
  18. Lines 228-232: Why do they suppose seasonal effects for S. pyogens and S. aureus? Support with bibliography.
  19. Line 246: Streptococcus in italics.
  20. Line 247: Must say microbiota.
  21. Line 264: You cannot conclude this since you do not present results in this regard made in this work
  22. References: Very poor. You must put them properly, put them in alphabetical order, the name of the journals must be written homogeneously, use the same typeface. Putting more care into references is also an important part of the manuscript.

Author Response

                                                                                    Pilsen, 27th May, 2020

Dear Reviewer 2,

Thank you for your comments to the article: "Epidemiological and Microbiological Aspects of The Peritonsillar Abscess

We went carefully through your comments. Please, find below our answers to each comment.

We also improved the level of English. Article was checked by a native speaker with biology education, focused on microbiology.

We believe that all the changes done increased the level of the article.

Kind regards,

David Slouka

Reviewer 3 Report

Topic is not original but of cliinical interest

Number of patients examined is considerable 

English has to be rewieved

The discussion paragraph is too fragmentary to interpret

Authors do not cite recent acquisitions on the role of microbiota in tonsillar infections

Therapeutic indications derived from the results obtained by the authors could be useful in clinical practice

Author Response

                                                                                  Pilsen, 27th May, 2020

Dear Reviewer 3,

Thank you for your comments to the article: "Epidemiological and Microbiological Aspects of The Peritonsillar Abscess

We went carefully through your comments. Please, find below our answers to each comment. Our answers are written in blue.

We also improved the level of English. Article was checked by a native speaker with biology education, focused on microbiology.

We believe that all the changes done increased the level of the article.

Kind regards,

David Slouka

Round 2

Reviewer 2 Report

Line 64: Say that people with and without antibiotic treatment were included.

Line 69: Say the explication about the three groups in the text.

Line 87: Put oxacillin-sensitivity test reference.

Line 89: Indicate that the bacterial species were determined by sequencing the 16S rRNA gene following the methodology of Aas et al

Author Response

                      Pilsen, 1st June, 2020

Dear Reviewer 2,

Thank you for your Comments and Suggestions to the article: "Epidemiological and Microbiological Aspects of The Peritonsillar Abscess

We went carefully through your comments and suggestions. Please, find below our answers to each comment. Our answers are written in red and in the text marked in yellow.

We believe that all the changes done increased the level of the article.

Kind regards,

David Slouka

Reviewer 3 Report

The Authors answered sufficiently to my questions

Author Response

Thank you very much.